# Low-dose CT Enhancement Network with a Perceptual Loss Function in the Spatial Frequency and Image Domains

**Kevin J. Chung**[1,2]                                                    JCHUN269@UWO.CA
**Roberto Souza**[3,4]                                     ROBERTO.MEDEIROSDESO@UCALGARY.CA
**Richard Frayne**[3,4]                                                RFRAYNE@UCALGARY.CA
**Ting-Yim Lee**[1,2]                                                    TLEE@ROBARTS.CA

[1] *Department of Medical Biophysics, University of Western Ontario, London, ON, Canada*

[2] *Robarts Research Institute and Lawson Health Research Institute, London, ON, Canada*

[3] *Departments of Radiology and Clinical Neuroscience, Hotchkiss Brain Institute, University of Calgary, Calgary, AB, Canada*

[4] *Seaman Family MR Research Centre, Foothills Medical Centre, Calgary, AB, Canada*

**Editors:** Accepted for MIDL 2020

## Abstract

We propose a dual-domain cascade of U-nets (*i.e.* a "W-net") operating in both the spatial frequency and image domains to enhance low-dose CT (LDCT) images without the need for proprietary x-ray projection data. The central slice theorem motivated the use of the spatial frequency domain in place of the raw sinogram. Data were obtained from the AAPM Low-dose Grand Challenge. A combination of Fourier space (F) and/or image domain (I) U-nets and W-nets were trained with a multi-scale structural similarity and mean absolute error loss function to denoise filtered back projected (FBP) LDCT images while maintaining perceptual features important for diagnostic accuracy. Deep learning enhancements were superior to FBP LDCT images in quantitative and qualitative performance with the dual-domain W-nets outperforming single-domain U-net cascades. Our results suggest that spatial frequency learning in conjunction with image-domain processing can produce superior LDCT enhancement than image-domain-only networks.

**Keywords:** Low-dose CT image enhancement, convolutional neural networks, dual-domain deep learning.

## 1. Introduction

Minimizing ionizing radiation dose in a CT scan is a major focus of imaging research. Reducing the tube current-exposure product (mAs) or the tube potential (kV) can lower the scan dose but introduces additional noise when reconstructing by analytical filtered back projection (FBP). Refining the acquisition signal (*i.e.* x-ray projection or sinogram) and image domain representations by iterative (Willemink et al., 2013) or deep learning reconstruction (McCann et al., 2017) can reduce noise and potentially improve diagnostic accuracy. Projection data, however, is often vendor proprietary and challenging to accurately synthesize from FBPs in an end-to-end network (Yin et al., 2019). By the central slice theorem, the spatial frequency of a projection line in the sinogram is encoded in the 2D Fourier Transform (FT) of the sinogram's FBP. We propose that restoring the spatial frequency of the CT image can alternatively refine the sinogram to enhance low-dose CT

(LDCT) images without requiring direct access to raw projection data. A cascade of spatial frequency and image-domain U-nets (Ronneberger et al., 2015) demonstrates this principle.

## 2. Materials and Methods

Our work derives from the use of dual-domain cascade of U-nets (*i.e.*, a W-net, operating in the spatial frequency ($k$-space) and image domains) for accelerated MR image reconstruction (Souza et al., 2019). As a baseline benchmark, we trained an image-domain U-net (I U-net) and W-net (II W-net). Four additional combinations of networks were trained that operated in the Fourier space and/or the image domain: spatial frequency-only networks (F U-net and FF W-net) and dual-domain W-nets in which the Fourier space was processed before the image-domain (FI W-net) or after (IF W-net). Dual-domain networks were connected by a Fourier transform between each U-net. The real and imaginary components of the complex spatial frequency signal were concatenated as two separate channels prior to input into a Fourier space network. Networks were trained end-to-end for 30 epochs using the Adam optimizer (Kingma and Ba, 2014) with a mini-batch size of 4. On-the-fly data augmentation simulated additional training data by translating, rotating, and reflecting the training images to minimize overfitting.

Noise and granular detail are key features of CT images. To that end, we used a modified perceptual loss function proposed by Zhao et al. (2017):

$$\mathbf{L} = \alpha K \mathbf{L}^{MS-SSIM} + (1 - \alpha)\mathbf{L}^{l_1} \tag{1}$$

where $\mathbf{L}^{MS-SSIM}$ is the multi-scale structural similarity (MS-SSIM) loss, $\mathbf{L}^{l_1}$ is the mean absolute error, $\alpha = 0.84$, $K = 1$ in the image domain, and $K = 2 \times 10^6$ in the spatial frequency domain. Constants were chosen such that the contributions of the two losses were balanced. We defer the specific definitions of $\mathbf{L}^{MS-SSIM}$ and $\mathbf{L}^{l_1}$ to Zhao et al. (2017). Losses were calculated in the respective domains in which the U-nets operated.

Paired sets of low-dose and routine-dose CT images were obtained from the AAPM Low-dose Grand Challenge (McCollough et al., 2017) and comprised of 10 patients with contrast-enhanced abdominal CT. Each patient image was provided at 1 mm and 3 mm slice thickness and both were included in our dataset as a measure of data augmentation. Five patient image sets were reserved for training ($4,748$ axial slices), two for validation ($1,193$ axial slices), and the remaining three for testing ($2,373$ axial slices). Image intensities were normalized to lie between 0 and 1 prior to input to the networks.

## 3. Results and Discussion

Networks involving an image domain U-net produced smoothed reconstructions while spatial frequency-only networks generated images with coarse texture features (Figure 1). The best networks (FI and IF W-nets) involved an image-domain U-net, and despite the prevalence of noise in the LDCT image, each of these networks demonstrated exceptional contrast between the contrast-enhanced vessels and liver tissue. The same level of contrast was not observed in the spatial frequency-only networks as the coarse texture impeded interpretability.

All networks outperformed LDCT in terms of quantitative metrics (Table 1). Friedman groupwise comparisons were significant ($p < 0.001$) and post-hoc pairwise comparisons with

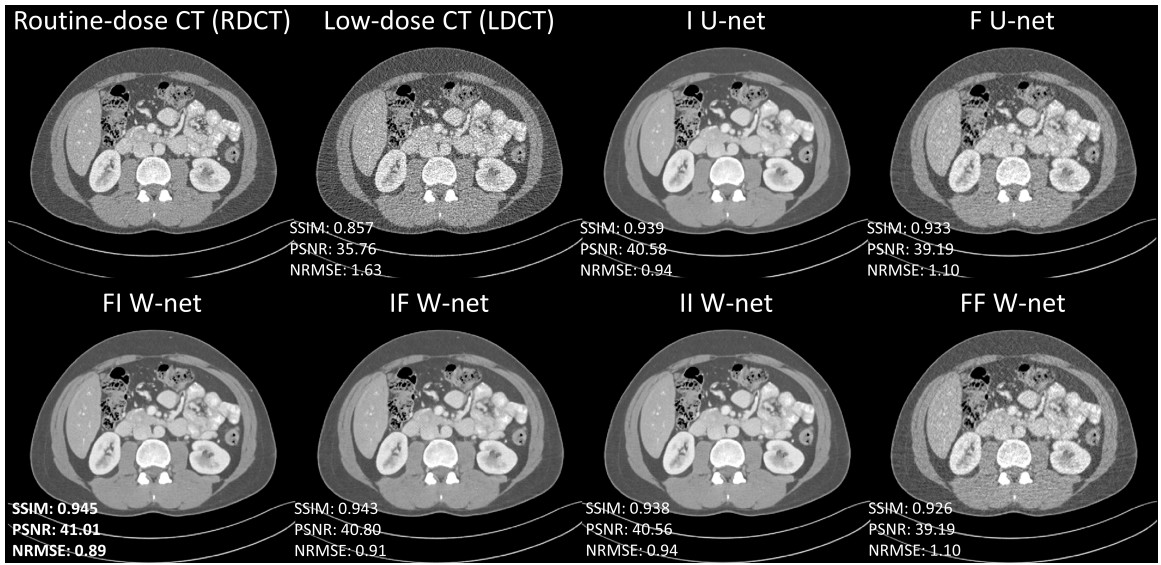

Figure 1: Sample enhancement of a low-dose axial slice using each network configuration.

Bonferroni correction indicated that all pairs except same-domain permutations (*i.e.*, I and II; F and FF; FI and IF networks) demonstrated statistically significant differences. The performance of the FI and IF W-nets exceeded the image-domain-only networks, validating the added benefit of operating in the spatial frequency domain with CT images. However, fine line patterns are visible in IF W-net reconstructed images (Figure 1) suggesting that despite statistical testing on performance metrics showing no difference, the FI W-net reconstructions may be qualitatively superior. These line patterns are likely due to uncorrelated high spatial frequency signal being restored by the Fourier space networks that does not necessarily correspond to pleasing perceptual performance.

Table 1: Quantitative performance metrics for each proposed enhancement approach. Mean ± standard deviation is reported. Best results are emboldened.

| Reconstruction | # Parameters | SSIM | PSNR (dB) | NRMSE (%) |
|---|---|---|---|---|
| FBP LDCT | - | 0.899±0.047 | 38.62±3.08 | 1.24±0.40 |
| I U-net | 11,690,753 | 0.957±0.020 | 43.00±2.73 | 0.74±0.21 |
| F U-net | 11,691,394 | 0.942±0.024 | 41.23±2.53 | 0.90±0.25 |
| II W-net | 23,381,506 | 0.957±0.021 | 43.05±2.83 | 0.74±0.22 |
| FF W-net | 23,382,788 | 0.943±0.023 | 41.31±2.47 | 0.89±0.24 |
| FI W-net | 23,382,147 | **0.961±0.019** | **43.39±2.68** | **0.71±0.20** |
| IF W-net | 23,382,147 | **0.960±0.019** | **43.30±2.68** | **0.72±0.20** |

Our initial study findings indicated that refining the spatial frequency of the CT image improves low-dose reconstructions when used in conjunction with an image-domain network.

Future work will involve comparing our proposed method to networks that directly operate on projection data.

## Acknowledgments

The authors would like to thank Dr. Cynthia McCollough, the Mayo Clinic, the American Association of Physicists in Medicine (AAPM), and grants EB017095 and EB017185 from the National Institute of Biomedical Imaging and Bioengineering, for access to the Low-dose CT Grand Challenge dataset. The authors would also like to thank the Amazon Web Services Cloud Credits for Research Program for access to cloud computing services. R.S. was supported by the T. Chen Fong Fellowship in Medical Imaging from the University of Calgary. R.F. holds the Hopewell Professorship of Brain Imaging at the University of Calgary.

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
