# OpenReview forum: "Low-dose CT Enhancement Network with a Perceptual Loss Function in the Spatial Frequency and Image Domains"
_MIDL.io/2020/Conference — MIDL 2020_

### Official Review · AnonReviewer4 · 2020-02-24
**interesting work, clearly presented**

**Rating:** 4
**Confidence:** 3

**Review:**

This paper tests the hypothesis that a dual-domain cascade of U-nets outperforms single-domain cascades. The results suggest that this is the case. The paper is straight forward, well-structured and the aims, methods and results and discussion are interesting, informative and clearly presented.

Minor comments:
url Link to Ronneberger’s unet paper is broken.

---

### Official Review · AnonReviewer2 · 2020-03-09
**Advantage over image-based U-net not demonstrated**

**Rating:** 1
**Confidence:** 4

**Review:**

The authors propose to perform a dual U-net in both frequency and image domain to denoise low-dose CT images.
Although the methodology is sound, I disagree with the authors in that "refining the spatial frequency of the CT image
improves low-dose reconstructions when used in conjunction with an image-domain network". PSNR improvement over a simple image-based U-net is marginal (+ 0.3 dB) especially when considered the additional complexity.
I am therefore sorry to recommend rejection.

---

### Official Review · AnonReviewer3 · 2020-03-11
**Easy to follow method with slightly unsatifilatory comparison**

**Rating:** 3
**Confidence:** 4

**Review:**

The paper proposed to denoise low-dose CT in both image and spatial frequency domain with the combination of L1 and MS-SSIM loss.

Pros:
-Well motivated
-The method is easy to follow

Cons:
-The proposed work only compared vertically with different compositon of the I and F Unet but has not compared horizontally with other LDCT denoising methods. For example: Kang, Eunhee, Junhong Min, and Jong Chul Ye. "A deep convolutional neural network using directional wavelets for low‐dose X‐ray CT reconstruction." Medical physics 44.10 (2017): e360-e375., which also proposed to denoise in the frequency domain
-Why K was set to 2x10^6 to overweight MS-SSIM loss for the frequency domain network?

---

### Official Review · AnonReviewer1 · 2020-03-12
**Interesting idea, some work to be done**

**Rating:** 3
**Confidence:** 5

**Review:**

This paper presents a method to denoise low-dose CT images. In contrast to previously proposed methods, no proprietary projection data is required. Instead, the method operates on the image domain as well as on the spatial frequency domain. The authors show that the combination of networks operating in the image and spatial frequency domain leads to quantitatively better denoising results.

Strengths
-	It’s an interesting idea to apply a U-Net not only in the image domain but also in the Fourier domain. It’s good that the method does not require sinogram data.
-	Experiments are well-structured and results are compared with statistical analysis.
-	The results show that operating in the spatial frequency domain has added value over operating only in the image domain.

Weaknesses
-	There has already been a lot of work on deep learning-based CT image denoising. E.g. using wavelet transforms instead of Fourier transforms https://aapm.onlinelibrary.wiley.com/doi/full/10.1002/mp.12344 or using generative adversarial networks https://ieeexplore.ieee.org/document/7934380. In this context, the use of a perceptual loss is also not novel https://ieeexplore.ieee.org/document/8340157. None of these works are mentioned in the paper.
-	The data set used is quite small and the denoising results are only evaluated using quantitative measures that don’t take into account for which clinical application images are made. It would be good to add evaluation using a clinical task, e.g. nodule detection.
-	The authors write that networks ‘demonstrated exceptional contrast between [..] vessels and liver tissue’ but this is not quantified in any way. In fact, all results in Fig. 1 look nearly identical, I’m not convinced that adding a spatial frequency domain network has much practical value.
-	Networks now operate in sequence, but it may be more interesting to operate them in parallel so that errors are not propagated.

Detailed comments
-	A method to denoise CT images in a non-image domain has previously been proposed:
-	Please explain how image intensities were normalized, was this by linear scaling between two HU values?
-	How was a value of 0.84 selected for alpha?

---

### Meta-Review · Area_Chair1 · 2020-04-05
**MetaReview of Paper59 by AreaChair1**

**Rating:** 3

**Metareview:**

Interesting idea, well written paper. I suggest the authors to include and compare with relevant previous work as suggested by the reviewers.

**Paper Type:**

methodological development

---

### Decision · Program_Chairs · 2020-04-11

Accept